# Qualitative community stability determines parasite establishment and richness in estuarine marshes

Tavis K. Anderson[1,2] and Michael V.K. Sukhdeo[3]

[1] Graduate Program in Ecology and Evolution, Rutgers University, New Brunswick, NJ, USA
[2] Virus and Prion Research Unit, National Animal Disease Center, USDA-ARS, Ames, IA, USA
[3] Department of Ecology, Evolution, and Natural Resources, Rutgers University, New Brunswick, NJ, USA

## ABSTRACT

The establishment of parasites with complex life cycles is generally thought to be regulated by free-living species richness and the stability of local ecological interactions. In this study, we test the prediction that stable host communities are prerequisite for the establishment of complex multi-host parasite life cycles. The colonization of naïve killifish, *Fundulus heteroclitus*, by parasites was investigated in 4 salt marsh sites that differed in time since major ecological restoration, and which provided a gradient in free-living species richness. The richness of the parasite community, and the rate at which parasite species accumulated in the killifish, were similar between the low diversity unrestored site and the two high diversity (10- and 20-year) restored marsh sites. The parasite community in the newly restored marsh (0 year) included only directly-transmitted parasite species. To explain the paradox of a low diversity, highly invaded salt marsh (unrestored) having the same parasite community as highly diverse restored marsh sites (10 and 20 yrs) we assessed qualitative community stability. We find a significant correlation between system stability and parasite species richness. These data suggest a role for local stability in parasite community assembly, and support the idea that stable trophic relationships are required for the persistence of complex parasite life cycles.

# INTRODUCTION

Parasites are considered to be ubiquitous components of ecosystems and the trophic strategy may represent more than 50% of potential interactions in food webs (*Price, 1980*; *de Meeûs & Renaud, 2002*). This pervasive nature is reflected in their many roles in the regulation of competitive and predator/prey interactions (*Hatcher, Dick & Dunn, 2006*), community structure (*Dobson & Hudson, 1986*; *Wood et al., 2007*), and ecosystem energy flow (*Mouritsen & Jensen, 1994*; *Kuris et al., 2008*). Additionally, there is a growing body of evidence which suggests that parasite species diversity is positively related to ecosystem functioning (*Thomas, Guegan & Renaud, 2005*; *Collinge & Ray, 2006*). Indeed, recent theoretical work has suggested that stable communities may require a balance of

Corresponding author
Tavis K. Anderson,
tavis.anderson@ars.usda.gov

antagonistic interactions (e.g., parasitism) with mutualistic interactions to achieve stability (*Mougi & Kondoh, 2012*). The underlying principle being that parasite life strategies are integrally coupled with populations of hosts, and that these hosts are critical not only for the parasites, but also to the dynamics of the free-living ecosystem (*Poulin, 2007*).

The reliance of parasite species on free-living hosts for establishment and persistence is particularly true for those parasites with complex life cycles. Free-living hosts are essential habitats for the parasites, and also serve as a route of transmission via trophic interactions (*Marcogliese & Cone, 1997*; *Marcogliese, 2002*). This co-dependence is reflected in host-parasite dynamics (*Grenfell & Dobson, 1995*; *Poulin & Thomas, 1999*), and has allowed the use of parasites as biological tags (*Criscione, Cooper & Blouin, 2006*) and as a metric of ecosystem restoration success (*Huspeni & Lafferty, 2004*). Indeed, it is generally accepted that functioning ecosystems with robust structured organizations tend to be rich in parasites (*Hudson, Dobson & Lafferty, 2006*; *Dobson et al., 2008*). It follows, therefore, that the stability of free-living populations and the free-living community, should be reflected in the dynamics of the parasite community (reviewed in *Lafferty et al., 2008*). However, outside of a handful of studies using topological analyses to investigate parasite diversity (e.g., *Chen et al., 2008*; *Anderson & Sukhdeo, 2011*), very few studies have considered the impact of free-living community dynamics upon the parasite community. Thus, there is scant empirical evidence to support the notion that the stability of the free-living community is a requirement in parasite establishment, or for the persistence of parasite communities.

Parasites have high host fidelity, and there should be a correlation between the success of establishment in a specific host population and the local stability of that community. Logically, a community that is locally stable (*May, 1972*; *May, 1973a*; *Ives & Carpenter, 2007*; *Allesina & Tang, 2012*) represents a predictable resource for complex life cycle parasites to establish in or upon. The standard approach to quantifying local stability has been to generate community matrices, and determine the real part of the largest eigenvalue; if this value is negative the system is considered stable (*May, 1972*). The generation of a community matrix, however, is challenging (i.e., it is difficult to measure interaction strengths to parameterize the community matrix), and alternate strategies have been developed that are based on the pattern of signs rather than their magnitude (*May, 1973b*; *Allesina & Pascual, 2008*). This approach is particularly tractable given the preponderance of topology-based food webs; the food web adjacency matrix is assigned random interaction coefficients (i.e., transformed to a community matrix), the dominant eigenvalue is calculated, and the process iterated (*Allesina & Pascual, 2008*). In doing so, qualitative comparisons of local stability are possible (*May, 1973b*; *Allesina & Pascual, 2008*). Specifically, a community that is more qualitatively stable is stable proportionally more often than systems that are not.

To assess the relationship between community stability and parasite establishment, we conducted a natural field experiment documenting parasite colonization of a naïve focal fish species within four salt marshes of differing local stabilities. We asked: does qualitative stability have an impact on the establishment of complex parasite life cycles? We report that

complex life cycle parasites are present in all but the least stable food web, and propose that stability in the free-living trophic network constrains the establishment and persistence of complex life cycle parasites.

## MATERIALS AND METHODS

### Defining the study system

We conducted our study of parasite establishment within an area where over 90% of estuarine marshes are heavily impacted due to decades of anthropogenic disturbances. Recent large-scale restoration projects with the goal to recreate 'pristine' New-England type salt marshes, enhancing and creating a variety of marsh habitats for wildlife, have created spatially delineated habitats reflecting a gradient in time since restoration: Mill Creek Marsh (20 yr: 40° 47′ 45″ N, 74° 02′ 30″ W), Harrier Meadow (10 yr: 40° 47′ 12″ N, 74° 07′ 3″ W), Secaucus High School Marsh (0 yr: 40° 48′ 17″ N, 74° 02′ 52″ W), and Oritani Marsh (unrestored: 40° 47′ 57″ N, 74° 05′ 07″ W). Data on the free-living community and the construction of trophic food web networks has been previously described (*Anderson, 2009*; *Anderson & Sukhdeo, 2011*).

### Field collections

We use the parasite community of a focal species, the common marsh killifish *Fundulus heteroclitus*, to assess the impact of community stability on parasite establishment. In our marshes, the host community (benthic invertebrates, birds, fishes) potentially transmit 11 species of helminth parasites to the common killifish (*Anderson & Sukhdeo, 2010*; *Anderson & Sukhdeo, 2011*; *Anderson & Sukhdeo, 2013*). We collected approximately 2000 fish from Kingsland Impoundment, a marsh area abutting our Harrier Meadow site. Upon capture, killifish were held in plastic cattle watering tanks (1.52 m in diameter and 0.61 m deep: containing ∼1000 L of water pumped from Kingsland Impoundment) for 7 days. The killifish were subjected to a series of anthelminthic treatments for 5 consecutive days, with the addition of metronidazole, praziquantel, levamisole and piperazine in standard concentrations (*Bishop, 2005*). Following treatment, fish were held in recovery tanks for 2 days. Prior to stocking field cages at each site, 30 fish ($n = 120$ total necropsied) were removed from the recovery tank and dissected to ensure the treatment process was effective and establish a time zero parasite community value.

Three cages (2.4 m long, 2.4 m wide, and 1.5 m deep) were placed in each of the four marsh sites. Each cage was made of plastic-coated galvanized wire panels with 1 cm mesh, this allowed for water flow and benthic macroinvertebrate colonization but excluded predators. Cages were placed in marsh sites two weeks prior to stocking. Following our fish treatment and recovery regime, we stocked each cage with 150 killifish. The resulting density (17 fish/m$^3$) was within the natural variation exhibited by local killifish populations. We placed the cages in water that ranged from 1.0 m–1.5 m in depth; there was no halocline in the range in which the cages were placed. To eliminate the possibility that differences in cage placement or construction may affect the results, cage location within each marsh was determined randomly. Fish cages were sampled weekly for four

weeks; at each sampling date 10 fish were removed from each cage ($n = 30$ per site each week). After the first 4 weeks, cages were sampled every two weeks ($n = 30$ per site) until cages were exhausted. Given prohibitive labor and time restrictions, a subsample of 15 fish were necropsied immediately after collection at each site. The fish were placed in a buffered 300 mg/L solution of tricaine methanesulfonate (MS-222) until cessation of opercula movement, followed by pithing of the brain and spinal cord. The MS-222 solution, and all external surfaces were examined for ectoparasites using a stereomicroscope. The eyes and gills were removed and individually examined with each gill arch examined separately. The viscera were removed and internal body organs were examined using a stereomicroscope. All helminths were heat fixed and stored in 70% ethanol pending examination. Platyhelminths and acanthocephalans were stained with acetocarmine and mounted in Permount. Nematodes were fixed and stored in a mixture of 5% glycerol and 70% ethanol, and identified after approximately 2 wk. Helminth parasites collected during necropsy were identified using the keys of *Hoffman (1999)*, *Anderson, Chabaud & Willmott (1974)*, *Schell (1970)* and primary literature (*Harris & Vogelbein, 2006*).

Field collections were conducted under scientific permits issued by the New Jersey Department of Environmental Protection, Division of Fish and Wildlife, Marine Fisheries Administration (#0558, #0628, #0746) and Bureau of Freshwater Fisheries (#0536, #06008, #07019). Fish euthanasia was in accordance with the 2000 Report of the American Veterinary Medical Association Panel on Euthanasia and approved by The Animal Care and Facilities Committee at Rutgers University (#00-012).

## Analysis of parasite community establishment

To document parasite species richness, we used species accumulation curves (SACs) implemented using the bias-corrected Chao2 estimator in EstimateS 8.0.0 (*Colwell, 2009*). Sample order was randomized 100 times without replacement and mean species richness was estimated for each sample. We calculated the fit of the asymptotic Michaelis-Menten equation $y = ax/(b + x)$, where $y$ is observed richness, $x$ is sample size, $a$ is the asymptote or predicted richness and $b$ is a measure of the rate at which the curve approaches the asymptote. A Kruskal-Wallis one-way analysis of variance (ANOVA) was used on mean species richness to determine whether there were significant differences between sites; multiple pairwise comparison was conducted using Dunn's test.

## Community stability analyses

Food web stability was assessed using the framework provided by *May (1973a)* and *May (1973b)* and extended by *Allesina & Pascual (2008)*. It is based on the concept of food web matrices and the dynamics of species densities in the network. *May (1973a)* and *May (1973b)* demonstrated that the community may be described via a matrix of interactions, and that a system will be locally stable if the eigenvalues of that matrix have negative real parts. Our approach for generating community matrices, and describing their characteristic eigenvalues is described in the Supplemental Information.

We also calculated the connectance of our food web matrices where connectance ($C = L_o/S^2$) is the number of realized links ($L_o$) divided by the number of possible links ($S^2$).

Measured in this way, $C$ is the average fraction of species in a community consumed by the average species, i.e., when $C = 0$ no species consume each other and when $C = 1$ all species consume all other species and themselves. In order to allow for between network comparisons of connectance we normalized the data following the method of *Gilbert (2009)*:

$$C_{norm(S)} = \frac{C_{(S)} - C_{\min(S)}}{1 - C_{\min(S)}} \tag{1}$$

where $C_{norm(S)}$ = normalized connectance for $S$ species; $C_{(S)}$ = connectance for $S$ species; and $C_{\min(S)}$ = the minimum value of connectance for $S$ species.

## RESULTS

A total of 600 fish were studied: 15 fish were necropsied from each of the collections (a subsample from the 30 fish collected) that occurred at each of the 4 sites over 10 samples, covering 14 weeks during the summer of 2008. Eight taxa of metazoan parasites were identified including nematodes *Dichelyne bullocki* and *Contracaecum* sp.; the digenean metacercaria of *Ascocotyle diminuta* and *Posthodiplostomum minimum*; monogeneans *Fundulotrema prolongis* and *Swingleus ancistrus*; the acanthocephalan *Paratenuisentis ambiguous*; the copepod *Ergasilus funduli*; these taxa infected more than 70% of the killifish examined. Parasite intensity per host ranged from 1 to 2146 for the directly transmitted monogeneans and copepod: parasite intensity per host ranged from 1 to 806 for the complex life cycle acanthocephalans, nematodes and digeneans.

The Oritani marsh (unrestored) included 71 species and had 5041 potential links of which 629 were realized, resulting in a normalized connectance of 0.113. The Secaucus Marsh (0 year) included 87 species and had 7569 potential links of which 627 were realized, resulting in a normalized connectance of 0.072. The restored marshes, Harrier Marsh (10 year) and Mill Creek (20 year) included 112 and 122 species respectively; the resulting values of normalized connectance were 0.088 for Harrier Marsh and 0.117 for Mill Creek Marsh. The estimated parameters for each marsh are summarized in Table 1.

Secaucus Marsh (0 year) had a parasite community consisting only of the directly transmitted monogenean *S. ancistrus* and the copepod *E. funduli* (Fig. 1). The remaining three marsh sites recorded observations of all 8 species of metazoan parasites (direct and complex life cycle) discovered during this study: Oritani Marsh (unrestored), Harrier Marsh (10 year) and Mill Creek Marsh (20 year). The mean species richness and parasite community displayed significant pairwise correlations between Oritani Marsh, Harrier Marsh and Mill Creek Marsh (Table 2). The accumulation of parasite species in each marsh was consistent with the asymptotic Michaelis-Menten equation: a steep slope with an early asymptote (Fig. 1). The goodness-of-fit was adequate for: Oritani Marsh (Fig. 1: $df = 149$, $r^2 = 0.53$, $SS = 58.64$, $S_{yx} = 0.63$); Secaucus Marsh (Fig. 1: $df = 149$, $r^2 = 0.95$, $SS = 0.51$, $S_{yx} = 0.058$); Harrier Marsh (Fig. 1: $df = 149$, $r^2 = 0.93$, $SS = 7.86$, $S_{yx} = 0.23$); and Mill Creek Marsh (Fig. 1: $df = 149$, $r^2 = 0.76$, $SS = 20.31$, $S_{yx} = 0.37$). Overall mean species richness differed significantly between the four marsh sites (Fig. 1: Kruskal-Wallis one-way
**Table 1 Summary of food web stability metrics for each estuarine food web.** Statistics include species richness ($S$), potential links ($S^2$), observed links ($L_o$), linkage density ($d$), connectance ($C$), normalized connectance ($C_{norm}$), minimum fraction of eigenvalues with negative real part (min$E$), average fraction of eigenvalues with negative real part (mean$E$), and maximum fraction of eigenvalues with negative real part (max$E$).

| Parameters: | Oritani Marsh (unrestored) | Secaucus Marsh (0 year) | Harrier Marsh (10 year) | Mill Creek Marsh (20 year) |
|---|---|---|---|---|
| Number of species; $S$ | 71 | 87 | 112 | 122 |
| Potential no. of links; $S^2$ | 5041 | 7569 | 12544 | 14884 |
| Observed no. of links; $L_o$ | 629 | 627 | 1206 | 1846 |
| Linkage density; $d$ | 8.86 | 7.21 | 10.77 | 15 |
| Connectance; $C$ | 0.125 | 0.083 | 0.096 | 0.124 |
| Normalized connectance; $C_{norm}$ | 0.113 | 0.072 | 0.088 | 0.117 |
| MeanE | 0.839 | 0.771 | 0.851 | 0.803 |
| MaxE | 0.911 | 0.852 | 0.943 | 0.930 |
| MinE | 0.750 | 0.664 | 0.736 | 0.650 |

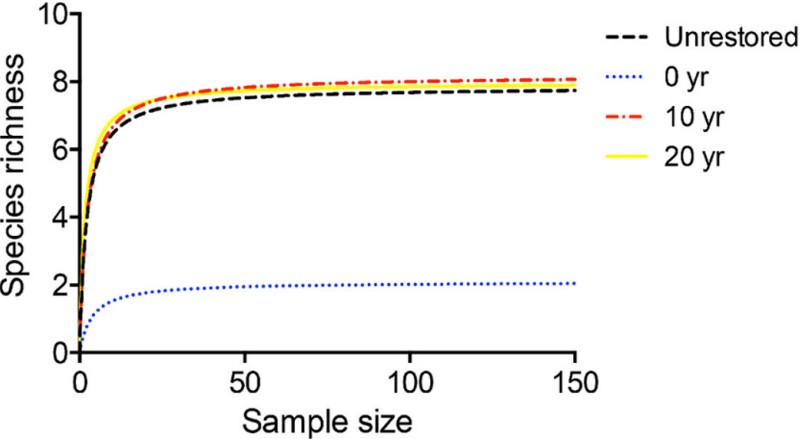

**Figure 1 Species accumulation curves documenting parasite establishment and richness.** Asymptotic randomized sample-based species accumulation curves for Oritani Marsh (Unrestored), Secaucus Marsh (0 year), Harrier Meadow Marsh (10 year) and Mill Creek Marsh (20 year). Curves represent the result of the bias-corrected Chao2 estimator of species richness based on *Fundulus heteroclitus* samples collected weekly from fish cages. For each curve, each point represents the mean of 100 estimates using randomized accumulation order.

ANOVA: $H = 350.5$, $df = 3$, $p < 0.0001$), with the mean species richness at Secaucus Marsh (0 year) significantly lower than the species richness observed at the other three sites (Fig. 1: Dunn's multiple comparison test: $p < 0.01$; Secaucus mean $= 1.1919 \pm 0.018$ SE; Oritani mean $= 7.447 \pm 0.055$ SE; Harrier mean $= 7.735 \pm 0.046$ SE; Mill Creek mean $= 7.662 \pm 0.034$ SE). There was no significant difference between the mean species richness observed at Oritani, Harrier and Mill Creek marshes (Fig. 1: Dunn's multiple comparison test: $p > 0.05$).

**Table 2 Parasite community richness correlations between 4 salt marsh food webs.** Pairwise Spearman nonparametric correlation coefficients on mean parasite species richness observed in the killifish, *Fundulus heteroclitus*, in each site.

| | Oritani Marsh (unrestored) | Secaucus Marsh (0 year) | Harrier Marsh (10 year) | Mill Creek Marsh (20 year) |
|---|---|---|---|---|
| Secaucus Marsh | 0.772 | | | |
| Harrier Marsh | 0.913[*] | 0.851 | | |
| Mill Creek Marsh | 0.935[*] | 0.822 | 0.957[*] | |

**Notes.**

[*] Represent significant correlation at $p < 0.05$.

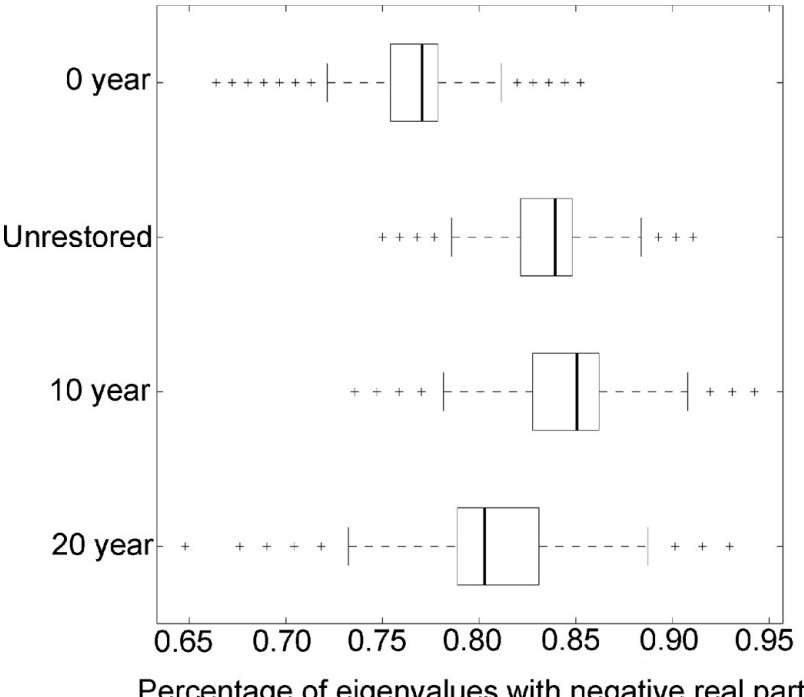

**Figure 2 Qualitative stability in 4 estuarine marshes in New Jersey, USA.** Percentage of eigenvalues with a negative real part out of 100,000 simulations for: Oritani Marsh (Unrestored); Secaucus Marsh (0 year); Harrier Marsh (10 year); Mill Creek Marsh (20 year).

For 100,000 randomizations of the community matrices generated from our four food webs, Fig. 2 shows the proportion of eigenvalues with a negative real part. None of the food webs appear to satisfy the criteria of *May (1973a)* and *May (1973b)* for stability: given the random interaction coefficients we imposed, no community matrix has 100% of the eigenvalues with negative real parts. However, there were significant differences in the mean fraction of eigenvalues with negative real parts between each of the marshes (One-way ANOVA with Tukey's pairwise comparison: all $p < 0.001$). The Secaucus Marsh (0 year) exhibits the lowest likelihood of achieving stability with a 77% mean fraction of eigenvalues with a negative real part (meanE), the maximum (maxE) and minimum (minE) fraction of eigenvalues attained for this food web were 78% and 75% respectively

(Table 1). Mill Creek Marsh, our oldest restoration (20 year) and highest diversity site, has a mean fraction of eigenvalues with negative real parts of 80.3%. Oritani Marsh (unrestored) has a mean fraction of eigenvalues with negative real parts of 83.9%. Harrier Marsh (10 year) has a mean fraction of eigenvalues with negative real parts of 85.1%. These data are aped by our normalized connectance values (Table 1), an oft-used metric in discussion of food web stability. Thus, Mill Creek Marsh, Oritani Marsh, and Harrier Marsh are more qualitatively stable than Secaucus Marsh. Using Spearman rank correlation, we document a strong relationship between qualitative stability and parasite richness (Spearman's rho = 0.8).

## DISCUSSION

Stable, predictable communities of organisms are thought to provide the foundational resource for parasites with complex life cycles (*Combes, 2001*; *Poulin, 2007*). The rationale being that in order to increase transmission, parasite life strategies become integrally coupled with host species and rely on stable populations of hosts. Indeed, the interaction between the parasite and its host is so vital that the death or absence of that host will result in the death of the parasite. A logical extension of this premise is that parasite species that rely on trophic interactions for transmission will only be present when stable predator-prey trophic links exist in ecosystems (*Marcogliese, 2002*; *Anderson & Sukhdeo, 2010*; *Anderson & Sukhdeo, 2011*; *Anderson & Sukhdeo, 2013*). Our data appears to support this thesis, our lowest diversity site (unrestored marsh) achieved the same parasite species richness as the more pristine, highly diverse marsh sites (10 and 20 year marshes): these three sites had similar stability, as measured by the fraction of eigenvalues that had negative real parts and normalized connectance. The deviation in parasite species richness occurred in our 0 year marsh site. This site was only colonized by directly transmitted parasites, and had a significantly lower fraction of eigenvalues with negative real parts and a lower value of connectance. The range of stability documented across our food webs suggests that there may be a critical threshold in the stability of host communities that is required for the persistence of complex life cycle parasites.

Our sample sizes constrain our inferences, but our proposition is plausible given that stable trophic interactions provide the critical connections allowing for trophic transmission of parasites, and the apparent challenge of predicting parasite diversity at large spatial scales. Indeed, *Dobson et al. (2008)* highlighted the difficulty in making predictions about parasite diversity patterns across geographical boundaries because of variation in the relationship between host diversity and parasite diversity. We suggest the reason for this difficulty is that the community metrics correlating with parasite diversity are not independent measures of system diversity, but also of the structure (cohesiveness) and stability of the food web, an inherently local phenomenon.

The concept that our parasite community is structured by local interactions is supported by the $\alpha-\beta$ diversity dichotomy demonstrated in SACs. This dichotomy links local and regional patterns of diversity (*Gering & Crist, 2002*) and allows for broad scale inferences about the mechanisms driving species richness patterns. Specifically, whether

the community is the result of local or regional processes, and whether community assembly is determined by niche or dispersal dynamics (*Gotelli & Colwell, 2001*; *Dove & Cribb, 2006*). In $\alpha$-dominated communities, richness is concentrated in individual samples, whereas $\beta$-dominated communities have dissimilar individual samples with most species richness existing as regional turnover among samples. These patterns are characterized by certain species accumulation curve (SAC) shapes: $\alpha$-dominated communities have a steep slope that rapidly reaches asymptote, and $\beta$-dominated communities have a more gradual slope and much later asymptotes. Our SAC data follows the shape exhibited by $\alpha$-dominated communities. This suggests interactive processes that exist on a local scale determine the parasite community in killifish in our system. Further, $\alpha$-dominated parasite communities are niche assembled, rather than being limited by dispersal (*Dove & Cribb, 2006*). Thus, it strongly supports the notion that the parasite species richness patterns in our focal species are the result of local processes such as the structure and stability of trophic interactions.

The view that parasites are attached to stable community interactions is slowly becoming a central theme in parasite ecology. This plays into a growing body of evidence that has begun to address whether there are critical clusters of interactions within free-living communities. Beginning with the work of *Gilbert (1977)* and *Gilbert (1979)*, and continued by *Thompson (2005)* and *Thompson (2009)*, it has been argued that the identification of interacting unit groups of species, in which natural selection acts upon all participants significantly, will likely provide considerable insight into community dynamics. These unit groups, termed 'stable evolutionary units,' are the result of co-adaptation and co-evolutionary processes based on their interactions (*Thompson, 1994*; *Thompson, 2005*). This argument proposes that two populations of natural enemies (or mutualists) can co-adapt and become dynamically stable over evolutionary time. Indeed, the existence of such stable two-species interactions within communities may also facilitate a third species (e.g., a parasite) that is able to adapt to, and exploit, the pre-existing interaction. For a complex life cycle parasite that relies upon trophic transmission, the initial predator/prey interaction provides the critical unit in its establishment and persistence. In essence, the structure of the host food web, exerts a strong selective pressure on the evolution of parasite transmission strategies and the subsequent patterns of parasite diversity observed in extant systems (*Marcogliese & Cone, 1997*; *Poulin, 2010*; *Anderson & Sukhdeo, 2011*).

A potential limitation of our study was our approach to analyzing the stability of the community without explicitly considering the effects of parasitism. There is a growing body of literature documenting that parasites affect host populations (*Hudson, Dobson & Newborn, 1998*; *Pedersen & Greives, 2008*), and host communities (*Wood et al., 2007*). Parasites may even function as the thread that binds food webs together (*Dobson, Lafferty, & Kuris, 2006*): a likely consequence of complex life cycle parasites introducing relatively weak interactions into 'long loops' (*Neutel, Heesterbeek & De Ruiter, 2002*). However, the effect of parasitism is not consistent across taxa: some parasitic species may have strong impacts on some species in their life cycle (e.g., *Lafferty & Morris, 1996*) but have weak or non-detectable impacts on others. This variation in effect is highlighted by theoretical

work that struggled to adequately characterize empirical host-parasite topological networks (e.g., *Petchey et al., 2008*). The approach we take to qualifying stability, by randomly generating community matrices, overcomes the challenge of modelling variation in parasite effects and reflects the stability of the entire system. Our approach captures and includes the effects of pathogens - including bacteria, prions, viruses, and parasites – on host population dynamics, without explicitly considering them. Given that we were primarily interested in predictors of parasite richness, our approach is adequate: we found that the local stability of free-living ecological communities is correlated with parasite establishment and richness. Although our approach is simple, our conclusion that stable ecological systems support richer parasite fauna is also intuitive. Clearly, future studies must address the dynamic properties of ecological networks to provide a more realised understanding of how parasites respond to the host community, and how the free-living community responds to parasitism.

## CONCLUSIONS

Our data suggests that as a result of the dependence of parasites upon their free-living hosts and the nature of the ecological network in which they reside, local stability can strongly influence parasite establishment. Though parasites permeate entire ecosystems – a position derived from the frequency of life cycles with one parasite species interacting with many free-living hosts – it is apparent that they are not always present, i.e., parasites do not appear to be everywhere all the time. Consequently, though parasites substantially alter common food web metrics (*Lafferty, Dobson & Kuris, 2006*), we would caution against inference using coarse indices of biodiversity or considering a parasite-host trophic link the equivalent of a predator-prey trophic link. Our approach posits that the structural patterns of trophic interactions in host-based food webs can illuminate patterns of parasite diversity and establishment (e.g., *Anderson & Sukhdeo, 2011*).

## ACKNOWLEDGEMENTS

We acknowledge the assistance of Jesse Stratowski in the construction, installation, and sampling of our experiment cages.

### Funding

This research was funded in part by a Willis A. Reid Jr. student research grant (The American Society of Parasitologists) to TKA and by the Rutgers University Foundation Parasite Research Fund to TKA and MVKS. The funders had no role in study design, data collection and analysis, decision to publish, or preparation of the manuscript.

### Grant Disclosures

The following grant information was disclosed by the authors:
The American Society of Parasitologists: Willis A. Reid Jr. student research grant.
Rutgers University Foundation Parasite Research Fund.

## Competing Interests

The authors have declared that no competing interests exist.

## Author Contributions

- Tavis K. Anderson conceived and designed the experiments, performed the experiments, analyzed the data, contributed reagents/materials/analysis tools, wrote the paper.
- Michael V.K. Sukhdeo contributed reagents/materials/analysis tools, wrote the paper.

## Animal Ethics

The following information was supplied relating to ethical approvals (i.e., approving body and any reference numbers):

Fish euthanasia was approved by The Animal Care and Facilities Committee at Rutgers University, Office of Research and Sponsored Programs (Protocol 00-012) in accordance with the 2000 Report of the American Veterinary Medical Association Panel on Euthanasia.

## Field Study Permissions

The following information was supplied relating to ethical approvals (i.e., approving body and any reference numbers):

Field collections were conducted under scientific permits issued by the New Jersey Department of Environmental Protection, Division of Fish and Wildlife, Marine Fisheries Administration (#0558, #0628, and #0746) and Bureau of Freshwater Fisheries (#0536, #06-008, and #07-019).

## Supplemental Information

Supplemental information for this article can be found online at http://dx.doi.org/10.7717/peerj.92.

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
