# Peer review of "Qualitative community stability determines parasite establishment and richness in estuarine marshes"

_PeerJ, doi:10.7717/peerj.92_

## Round 0.1 · original submission · Major Revisions

We have two reviews. Both find your paper interesting but both have a number of comments. You must make a revised version and the list of your response to each of these comments.

·

Basic reporting

(pp.2)
L.28-30: the sentence is not clear. The definition of "free-living ecosystem" is ambiguous and I didn't understand at first. Also note that the sentence is not completed, although I did understand what authors intended to mean. Similar mistakes appeared in the text several times. They need reconsider their basic grammar.

L.35: Authors have mentioned in L.33 the term “evolutionarily stable” transmission “strategy”, but the way in which this term is used is not appropriate. In the context of evolutionary theory, “evolutionarily stable strategy (ESS)” is often defined as a strategy satisfying the following property: if the system is occupied by the strategy (ESS), then no rare mutant with deviant strategy can invade the system. The cited authors (Marcogliese & Cone 1997; Marcogiese 2002) do not discuss in this context. If authors intended to assert the discussion about the dependence of ESS on the stability of ecosystems, then authors need to cite other references for discussing ES-transmission strategy. Throughout this article, authors use the term “stability” to characterize the system in ecological aspects and locally, in the same way as Robert M. May had discussed the ecosystem stability;

L. 36: parasite and host population dynamics -> host-parasite dynamics.
The epression “Host-parasite” is more often used than “parasite and host”; the latter is rarely used (see Springer Examplar).

L.39: functioning ecosystem, ->functioning ecosystem


(pp.2-3) L.39-49: the introduction of problems to solve. There are many sentences, but the logic is unclear so that readers are required to consider very carefully. Authors should also organize the importance of solving their question in an ecological context.

L.43-46: the sentence is not completed. Or, if you have meant to complete, you had better avoid using participial construction. It’s very hard to read, although I grasp what is intended to mean.

(pp.3-4) L.50-66: the introduction of the method with which authors analyze the ecosystem local stability.


(pp.7) L.149-: I suggest authors put the Community stability dynamics into appendix. The method used here is famous among community ecologists, but mathematically non-trained readers may feel tired of reading this part. In effect, linear stabilization is investigated and this explanation is not necessary in the main text.

(pp.9) L.182-: the definition of the linkage density (d) is unclear or non-referred, appearing only in Table 1.

(pp.10) L.204-: Authors cite Table 1 several times. I suggest they refer to Table 1 only once by adding, for example, “the estimated parameters for each study area are summarized in Table 1”.

(pp.12) L.254-: I suggest they avoid using participial construction; it yields the logic ambiguous and hard for me to read.

(pp.14) L.300 (Gilbert 1977; 1979)->Gilbert (1977; 1979), & (Thompson 2005; 2009)->Thompson (2005; 2009)

(pp.13) L.271, L.276 “that” is doubled or tripled in a single sentence. I suggest they write in more simplified manner.

In fig 1, the differences in curves are not clear. Authors need to use more vividly different curve styles, such as largely dotted, dashed, solid, and so on.

Authors have suggested the strong association between parasite richness and community stability. This focus is lacking in previous studies. I suggest authors cite: A. Mougi, M. Kondoh*. (2012) Diversity of interaction types and ecological community stability. (Science) 337:349-351., for the newest result (as far as I know) which is working on how the community stability is maintained in a trophic interaction network.

(pp.15) L.325 I suggest they drop “sensu”. The citation is incorrect: here this reference appeared the first time. Is this necessary? Or, discussion is insufficient? Also note that you describe as “Lafferty, et al. 2006b”

Experimental design

I suggest they prepare a schematic figure of the experimental protocol.

Validity of the findings

No comments

Additional comments

Comments on abstract:
“What was found” and “what was suggested from the result” are unclear. They write detailed materials & methods in the abstract, but their abstract is not informative. I suggest that they mention more about the background of this study as written in the Introduction part. Also I suggest that the title is indirect; they can institute more strongly about their interesting results.

Throughout this article description, their English is poor or not complete in grammar.

This work does play an important role for the further study of community stability theory. The investigation is done in a large scale so that all those who are interested in this area will focus on the aspects of parasite richness.

Reviewer 2 ·

Basic reporting

no comments

Experimental design

no comments

Validity of the findings

no comments

Additional comments

A role for community stability in parasite establishment in estuarine marshes
TK Anderson and MVK Sukhdeo

Dear authors,

This study investigated the stability of communities including parasites with complex life cycles. They constructed the population dynamics model and evaluated the community stability cased on the local stability analysis. They examined four natural communities which differ species diversity (host species diversity and parasite diversity). They found that the highly parasite diverse communities are more stable than communities with lower richness of parasites. This result is interesting and topical because recent studies are going to examine the roles of different types of species interaction to community stability. Before publication, however, they should clarify the method of population dynamics model and some points.
First, we cannot find the topology of community networks in this model. Second, we cannot find the way to consider the parasite in the model. Third, we cannot find the way to incorporate the complex life cycles in the model. Fourth, we cannot find the way to consider the conversion efficiencies in the model. Fifth, how did you determine the values of the intrinsic growth rates r ?
In addition, I am interested in the mechanism of the results. Why highly diversity of parasites tends to stabilize the system? Also, is this a general pattern? By controlling the proportion of parasites in the (model) communities, you can systematically examine the effects of parasite diversity on the community stability. I think that this is not difficult. If you consider this result, the paper will be greatly improved.

---

## Round 0.2 · Minor Revisions

Please make the final version considering this comments.

Reviewer 1 ·

Basic reporting

They found a correlation between the local stability of food webs and parasite diversity in four empirical systems. This is an interesting approach but the mechanism cannot be found because the parasite is not included in the stability analysis. Although in their response, they mentioned that the foodweb stability affects the parasite diversity, this is not possible because the parasite diversity and food web stability is independent in this approach. This is based on the assumption that the parasite does not influence the host. At least, they should add the potential explanation of filling in the gap.

Experimental design

no comments

Validity of the findings

no comments

---

## Round 0.3 · accepted · Accept

Thank you for addressing the previous concerns.